# Land Cover and Land Use Mapping of the East Asian Summer Monsoon Region from 1982 to 2015

Yaqian He [1,†], Jieun Oh [2,†], Eungul Lee [2,*] and Yaeone Kim [2]

1    Department of Geography, University of Central Arkansas, Conway, AR 72035, USA; yhe@uca.edu
2    Department of Geography, Kyung Hee University, Seoul 02447, Korea; dhwldms17@khu.ac.kr (J.O.);
     dpdnjs9808@khu.ac.kr (Y.K.)
*    Correspondence: eungul.lee@khu.ac.kr; Tel.: +82-2-961-9268
†    These authors contributed equally to this work.

**Abstract:** Owing to the recent intensification of the East Asian summer monsoon, the frequency of floods and dry spells, which commonly affect more than one billion people, is continuously increasing. Thus, understanding the causes of changes in the EASM is paramount. Land cover and land use change can perturb a regional climate system through biogeophysical and biogeochemical processes. However, due to the scarcity of temporally continuous land cover and land use maps, the impact of land cover and land use change on the EASM is still not thoroughly explored. In the present study, this limitation was addressed via the production of annual land cover and land use maps of the East Asian summer monsoon region covering a period of 34 years (1982–2015). This was achieved through a random forest classification of phenological information derived from the Advanced Very High-Resolution Radiometer Global Inventory Modeling and Mapping Studies Normalized Difference Vegetation Index dataset and terrain information from the Advanced Land Observing Satellite World 3D—30 m Digital Surface Model data. Nine ecological zones were involved in the random forest classification and the classified map in 2015 was validated using very high-resolution images obtained from Google Earth. The overall accuracy (73%) of the classification map surpasses the Moderate Resolution Imaging Spectroradiometer and Global Land Surface Satellite land cover products for the same year by ~7% and 4%, respectively. According to our classified maps, croplands and forests significantly increased in the East Asian summer monsoon region from 1982 to 2015. The dominant transition in these three decades was from croplands to forests.

**Keywords:** East Asian summer monsoon; phenology; Digital Surface Model; random forest; Normalized Difference Vegetation Index; land cover; land use classification

## 1. Introduction

The East Asian summer monsoon (EASM) is a prominent climate phenomenon that is associated with complex spatial and temporal characteristics [1]. This phenomenon impacts both the subtropics and midlatitudes, and the associated rainfall belt stretches across many countries including China, Korea, and Japan [1,2]. The EASM is an ecologically and socially important climate system [3]. Rainfall associated with the monsoon supports large-scale agricultural activities and forests that sustain the livelihood of more than a billion people [4]. The recent intensification of the EASM, with more frequent floods and intense dry spells, resulted in the 2018 Japan flood–heat wave succession event [5,6]. During this event, more than 1000 people lost their lives within a month and tremendous economic losses were also recorded [5,6]. Therefore, elucidating the causes of variations in the EASM is vital to improve understanding of its patterns, dynamics, and future impacts.

According to observational and modeling studies, multiple natural and anthropogenic drivers, including the orbital forcing, Himalayan–Tibetan Plateau uplift, intertropical convergence zone (ITCZ) movement, Atlantic meridional overturning circulation (AMOC)

teleconnections, aerosol emissions, and greenhouse gas effects, account for EASM variations [7–10]. Land cover and land use change (LCLUC), which can locally alter the exchange of energy and water between the land surface and atmosphere [11–14] and exert a significant influence on variations in the EASM [15,16]. Fu (2003), for instance, suggested that the destruction of vegetation can weaken the EASM [15]. A reduction in monsoon rainfall across the EASM region in response to past and future LCLUCs was also identified by Quesada et al. (2017), who used the model outputs from the Coupled Model Intercomparison Project Phase 5 (CMIP5) [17]. In contrast, based on the Weather Research and Forecasting (WRF) model, Zhao and Wu (2017) utilized satellite-derived land surface data to show that, based on the LCLUC between 1980 and 2010, rainfall associated with the monsoon increased in the south of the EASM region, but decreased in the north [18]. LCLUC also affects temperature in the EASM region. Niu et al. (2019) applied future LCLU data from the Land Use Harmonization (LUH) project into the 4th Regional Climate Model (RegCM4) and found LCLUC induced ~0.1–0.3° changes in surface air temperature [19]. However, most of these previous studies were based on a potential vegetation map [15], two static maps [18], or maps from a land use model [17,19]. These data were unsuitable for capturing actual year-to-year and long-term LCLU dynamics. For instance, the LUH data used in most climate models of the Coupled Model Intercomparison Phase 6 (CMIP6) and CMIP5 projects could not capture the long-term increase in forest coverage in China due to afforestation projects [20,21] (Figure A1). Thus, it may over- or under-estimate the associated effects on the EASM. Therefore, continuous, annual-scale LCLU data covering a long period is crucial for enhanced quantification of LCLUC effects on the EASM.

LCLU mapping in the EASM region has been attempted in several previous studies. Xiao et al. (2005), for example, utilized the Moderate Resolution Imaging Spectroradiometer (MODIS) data to map the cultivation of paddy rice in southern China for 2002 [22]. Sharma et al., (2016) utilized high-resolution imagery from the Landsat 8 Operational Land Imager to generate an LCLU map with a resolution of 30 m for Japan covering 2013–2015 [23]. Based on data from the same source, Piao et al. (2021) reported changes in the forest cover of North Korea using 18 LCLU maps from 2001 to 2018 [24], and Hansen et al. (2020) identified the tropical forest changes during same period [25]. In contrast, Seo et al. (2014) mapped the landscape in the catchment of Haean in South Korea and documented the LCLU types for 2009–2011 via annual field campaigns [26]. However, considering the restricted availability of MODIS data, cloud contamination of Landsat data and labor involved in field trips, maps for LCLU prior to the year 2000 were rarely produced in previous studies. Potapov et al. (2020) provided an annual-scale Landsat data source from 1997 to present for mapping LCLU, while they did not generate the land cover products [27]. Notably, Song et al. (2018) offered an annual, global vegetation continuous fields product for the time period, 1982 to 2016 [28]. He et al. (2017) employed the Advanced Very High-Resolution Radiometer (AVHRR) data from 1982 to 2013 to produce annual, continuous, and long-term LCLU maps of China [29]. Xu et al. (2020) then created annual land cover maps by integrating AVHRR, MODIS, and Landsat data for the period from 1980 to 2015 [30]. However, these maps involved only a few LCLU types [28] and part of the EASM region (e.g., China) [29]. The recently released long-term Global Land Surface Satellite-Global Land Cover (GLASS-GLC) product spans the entire EASM region [31], but the training and validation results derived from 30 m Finer resolution observation and monitoring of global land cover version 2 (FROM-GLC_v2) data and FLUXNET sites are based on the entire globe, which elevate uncertainties for the EASM region.

The objective of the present study is to produce and validate annual LCLU maps for the EASM region covering a duration of 34 years (1982–2015) to overcome existing limitations. The maps were generated via the random forest classification of phenological information derived from the AVHRR Global Inventory Modeling and Mapping Studies (GIMMS) Normalized Difference Vegetation Index (NDVI) dataset and terrain information from the Advanced Land Observing Satellite (ALOS) World 3D-30 m (AW3D30) Digital Surface Model (DSM). The LCLU maps, covering latitudes 100–146° E and longitudes 20–55° N

(Figure 1), include the east of China, east of Mongolia, southeast of Russia, the Korean Peninsula, and Japan. We validated the classification maps using very high-resolution images from Google Earth Pro and compared those with land cover datasets from the GLASS-GLC and Climate Change Initiative land cover (CCI LC) products. Based on the validated maps, spatiotemporal changes in LCLU for the EASM region over the last three decades were highlighted using linear regression and spatial transition analyses.

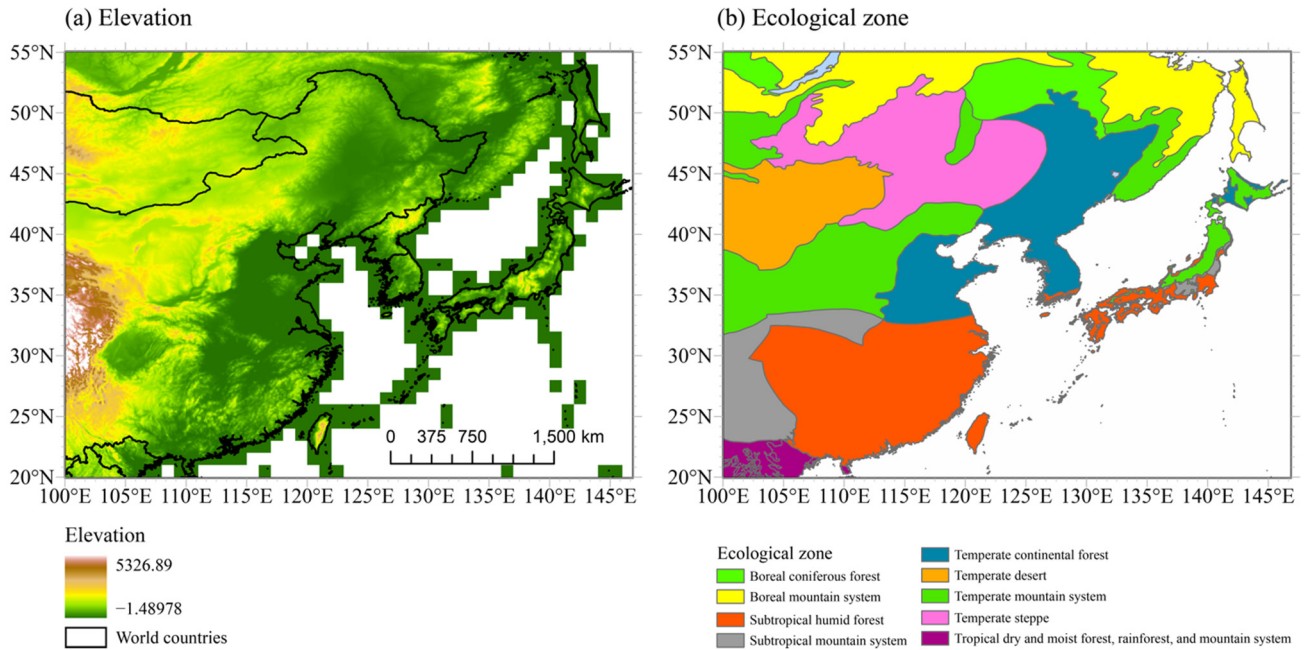

**Figure 1.** Study area of the EASM region, (**a**) elevations and (**b**) ecological zones.

## 2. Materials and Methods

### 2.1. Materials and Pre-Processing

The three groups of data used in this study are the following: (1) classification data from the third generation AVHRR GIMMS NDVI, AW3D30 DSM, and MODIS land cover (MCD12Q1) datasets; (2) validation data from very high-resolution Google Earth Pro imagery; and (3) inter-comparison data from the GLASS-GLC and European Space Agency (EAS) CCI LC datasets. These data and the associated pre-processing steps are described subsequently.

2.1.1. Classification Data

In the study, the second version of the third generation GIMMS NDVI (NDIV3g) data from 1982 to 2015 (http://poles.tpdc.ac.cn/en/data/9775f2b4-7370-4e5e-a537-3482c9a8 3d88/, [32], accessed on 15 November 2020) was utilized. The dataset was generated every 15 days using a spatial resolution of 1/12°. The NDVI, which reflects seasonal growth patterns of vegetation, is widely employed to distinguish LCLU types [33–35]. The second version of the NDVI3g data involves three values for the characterization of quality; that is, 0 for a good value, 1 for a value estimated via spline interpolation, and 2 for a value possibly influenced by snow/cloud cover. We retained the pixels with quality flag values of 0 or 1, while pixels with values of 2 were excluded. To minimize outliers in the NDVI values associated with atmospheric effects [36], the median filtering method in the TIMESAT software (https://web.nateko.lu.se/timesat/timesat.asp, accessed on 10 August 2020) [37,38] was used. We then employed a double logistic method in the software to smooth the NDVI time-series data, and this process was performed twice (i.e., the first involved one growing season parameter setting, while the other required two growing season parameter setting) to identify pixels associated with two growing seasons,

as reported in a previous study [29]. Pixels involving two growing seasons were then directly considered croplands because natural vegetation is linked to one growing season in the EASM region. Regarding pixels with one growing season, we applied R programming language to extract 19 phenological metrics from the smoothed NDVI time-series of each year from 1982 to 2015, such as the maximum NDVI, the minimum NDVI, and Julian day of the start of the season (Table 1), to enhance the land surface classification. We did not use TIMESAT, as it can only produce 13 phenological metrics. These phenological metrics were used as classification features in the random forest classifier.

**Table 1.** Classification features for the random forest classifier.

| Phenological Metrics | |
|---|---|
| 1 | Maximum NDVI value |
| 2 | Minimum NDVI value |
| 3 | Julian day of maximum NDVI value |
| 4 | Julian day of minimum NDVI value |
| 5 | Integral of NDVI between Day 105 and Day 315 |
| 6 | Integral under the NDVI curve |
| 7 | Maximum derivative of NDVI curve |
| 8 | Minimum derivative of NDVI curve |
| 9 | Julian day of maximum derivative of NDVI curve |
| 10 | Julian day of minimum derivative of NDVI curve |
| 11 | Julian day of start season |
| 12 | Julian day of end season |
| 13 | NDVI value of start season |
| 14 | NDVI value of end season |
| 15 | Integral between maximum derivative and minimum derivative |
| 16 | Integral between start season and maximum value |
| 17 | Integral between end season and maximum value |
| 18 | Maximum NDVI value—minimum NDVI value |
| 19 | Maximum NDVI value/integral under the NDVI curve |

The DSM data served as a secondary classification feature in the random forest classifier due to variations in the surface elevation across the EASM region (Figure 1a). The AW3D30 data utilized in the study is a global dataset generated using images collected from 2006 to 2011 via a panchromatic remote-sensing instrument for stereo mapping (PRISM) aboard the ALOS [39]. We obtained the latest 30 m version in 2021 (https://portal.opentopography.org/raster?opentopoID=OTALOS.112016.4326.2, assessed on 9 August 2021), and removed invalid pixels (i.e., cloud and snow pixels). To ensure consistency with the spatial resolution for the GIMMS NDVI3g data, the DSM data was resampled to a resolution of $1/12°$ using the bilinear method.

The annual Collection 6 MODIS MCD12Q1 land cover datasets covering the period from 2001 to 2010 (https://lpdaac.usgs.gov/products/mcd12q1v006/, assessed on 26 April 2021) served as a reference for the identification of training areas and the selection of class labels for the random forest classification. The MODIS MCD12Q1 collection, which has a spatial resolution of 500 m, was derived from the supervised classification of MODIS Terra and Aqua reflectance data [40]. Several classification schemes are used in the MCD12Q1, such as the International Geosphere-Biosphere Programme (IGBP) (17 classes), the University of Maryland (UMD) (16 classes), and the leaf area index (LAI) (11 classes). In the present study, we used the widely utilized IGBP classification scheme that involves 17 classes [26,41–43]. We aggregated 500 m to the spatial resolution of the GIMMS NDVI3g using the majority aggregation method [44]. Pixels that were unchanged from 2001 to 2010 in the EASM region were used as reference data, and these produced abundant training data (Table 2). The LCLU types derived from MODIS IGBP classification scheme for those unchanged pixels (i.e., reference data) were used as our classification classes (Table 2). We eliminated the classes containing few pixels (i.e., closed shrublands, open shrublands, permanent wetlands, and permanent snow and ice), because these were insufficient for

training of the random forest classifier. Therefore, although the MODIS IGBP scheme involves 17 classes, the maps produced are limited to 13 classes (Table 2).

**Table 2.** Number of pixels for each unchanged LCLU type in each ecological zone in the EASM region.

| Classified Class | Number of Pixels | | | | | | | | |
|---|---|---|---|---|---|---|---|---|---|
| | BC | BM | SH | SM | TS | TD | TC | TM | TP |
| Evergreen needleleaf forests | 51 | 584 | 193 | 229 | 5 | | 2 | 46 | |
| Evergreen broadleaf forests | | | 1546 | 192 | | | | | 1256 |
| Deciduous needleleaf forests | 327 | 903 | | | 32 | | | 6 | |
| Deciduous broadleaf forests | 288 | 277 | 125 | 634 | 68 | | 5088 | 2747 | |
| Mixed forests | 599 | 3641 | 1528 | 927 | 18 | | 638 | 2321 | 2 |
| Woody savannas | 4120 | 7600 | 7932 | 1711 | 484 | | 464 | 485 | 486 |
| Savannas | 690 | 1670 | 5380 | 2000 | 63 | | 1197 | 541 | 428 |
| Grasslands | 105 | 2318 | 10 | 1088 | 15,885 | 3041 | 203 | 12,757 | 19 |
| Croplands | 218 | 7 | 2550 | 488 | 1865 | 138 | 10,516 | 3156 | 175 |
| Urban and built-up lands | 10 | 9 | 832 | 28 | 11 | | 330 | 139 | 6 |
| Cropland/natural vegetation mosaic | | | 1647 | 37 | | | 14 | 45 | 38 |
| Barren | | 1 | 2 | 26 | 1 | 8191 | 22 | 214 | |
| Water bodies | 60 | 362 | 577 | 23 | 60 | | 294 | 124 | 23 |
| Total | 6471 | 17,511 | 22,344 | 7383 | 18,493 | 11,372 | 18,770 | 22,589 | 2436 |

BC: Boreal coniferous forest; BM: Boreal mountain system; SH: Subtropical humid forest; SM: Subtropical mountain system; TS: Temperate steppe; TD: Temperate desert; TC: Temperate continental forest; TM: Temperate mountain system; TP: Tropical dry and moist forest, rainforest and mountain system.

2.1.2. Validation Data

We collected the very high-resolution images obtained from Google Earth Pro in 2015 as validation data. Initially, we applied a stratified sampling scheme to randomly select 1896 sample points with at least 15 points for each class following [45], aiming a precision of 10% and a confidence level of 85%. The land cover types in $1/12° \times 1/12°$ squares centered in the sample points, representing pixels of the NDVI data, were then interpreted. The sample was assigned the dominant class in the square. Even though some squares involved mixtures, classes were determined based on the dominant LCLU types in their surroundings. Some sample points in 2015, however, were not associated with high-resolution images, and thus 2014 and 2016 imagery data were utilized. If the classes for 2014 and 2016 were consistent, those were assigned to sample points as the 2015 land cover types. Sample points without very high-resolution images from 2014 to 2016 were deleted, and this produced 413 points for validation. Due to the inability to differentiate subclasses sometimes, the evergreen broadleaf, deciduous needleleaf, deciduous broadleaf, and mixed forests, in addition to the woody savanna and savannas, were combined under the forest class. Consequently, the validation process has the following seven classes: forests, grasslands, croplands, urban and built-up, cropland/natural vegetation mosaic, barren, and water bodies.

2.1.3. Inter-Comparison Data

To further assess the reliability of the classification maps, two additional datasets were introduced for comparison. The first was the 1982–2015 GLASS-GLC data generated by Liu et al. (2020) using multiple data from GLASS climate data records (CDRs), such as the NDVI, leaf area index, and fraction of absorbed photosynthetically active radiation (FAPAR) [31]. The GLASS-GLC data has a spatial resolution of 5 km and comprises seven classes (https://doi.pangaea.de/10.1594/PANGAEA.913496, accessed on 8 January 2021). The second was annual CCI LC data from 1992 to 2015 (https://www.esa-landcover-cci.org, accessed on 8 January 2021). These CCI data were generated by ESA using sources such as the Medium Resolution Imaging Spectrometer (MERIS) and Système Probatoire d'Observation de la Terre Vegetation (SPOT-VGT) NDVI [46]. The spatial resolution of the CCI data was

300 m and these were associated with 22 classes. Both datasets were resampled using the majority aggregation method to produce a spatial resolution of $1/12°$, to ensure consistency with the NDVI3g data. Classes common to GLASS-GLC, CCI, and our classified maps based on descriptions were also created to enable comparison (Table 3).

**Table 3.** Common classes for inter-comparing GLASS-GLC, CCI, and our classified maps.

| Common Class | Classified Map | Glass-GLC | CCI LC |
| --- | --- | --- | --- |
| **Forests** | Evergreen needleleaf forests<br>Evergreen broadleaf forests<br>Deciduous needleleaf forests<br>Deciduous broadleaf forests<br><br>Mixed forests<br>Woody savannas<br>Savannas | Forest | Tree needle-leaved evergreen closed to open (>15%)<br>Tree broadleaved evergreen closed to open (>15%)<br>Tree needle-leaved deciduous closed to open (>15%)<br>Tree broadleaved deciduous closed to open (>15%)<br>Tree broadleaved deciduous closed (>40%)<br>Tree mixed<br>Mosaic tree and shrub<br>Mosaic herbaceous |
| | | Shrubland | Shrubland<br>Shrubland evergreen<br>Shrubland deciduous |
| **Grasslands** | Grasslands | Grassland<br>Tundra | Grassland |
| | | | Tree cover flooded fresh or brackish water<br>Tree cover flooded saline water<br>Shrub or herbaceous cover flooded |
| **Cultivated and managed vegetation/agriculture** | Croplands<br><br>Cropland/natural vegetation mosaic | Cropland | Cropland rainfed<br>Cropland rainfed herbaceous cover<br>Cropland irrigated<br>Mosaic cropland<br>Mosaic natural vegetation |
| **Urban and built-up** | Urban and built-up lands | | Urban |
| **Water** | Water bodies | Snow/Ice | Snow and ice<br>Water |
| **Barren** | Barren | Barren land | Bare areas<br>Bare areas unconsolidated<br>Sparse vegetation |

## 2.2. Methods

The analysis method includes the following main steps: (1) land surface classification and post-processing, (2) accuracy assessment, and (3) change detection. Considering that pixels involving two growing seasons were already assigned as croplands, the land surface classification step only focuses on pixels associated with one growing season.

### 2.2.1. Classification and Post-Processing

In the present study, we employed the widely used random forest classifier [47,48] to classify the LCLU types. The random forest classifier involves several classification trees, which vote to produce one outcome for each pixel [49]. Users of this classifier are required to define the following two parameters: the number of decision trees produced (*ntree*) and the number of variables available for splitting at each node (*mtry*). Based on a previous study [29] and experiments, 500 was chosen as the value for the *ntree* and the default value (i.e., the square root of the number of predictor variables) for *mtry*. The 19 phenological metrics and DSM associated with pixels that exhibited unchanged LCLU types based on the MODIS land cover data served as the reference data (Equation (1)). These reference data were randomly split, and 25% was used to train (training data) the random forest classifier, while the remaining 75% served as the validation data to evaluate the performance of

the classifier (Figure 2). Considering the vast spatial extent of the EASM region and that phenological metrics may differ from area-to-area, even for the same LCLU type, we partitioned the EASM region into nine zones based on the ecological zones advanced by the Food and Agriculture Organization (FAO) (https://www.fao.org/forestry/fra/80298/en/, accessed on 22 July 2021; Figure 1b). The classification was conducted for each ecological zone using the R random forest package [50].

$$LCLU_{type} = f_{random\ forest}\left(x_{phe},\ x_{ele}\right) \tag{1}$$

where $LCLU_{type}$ is the classes of the unchanged LCLU pixels from MODIS land cover data (i.e., training data), $x_{phe}$ and $x_{ele}$ are the corresponding 19 phenological metrics and DSM values of those training data, respectively.

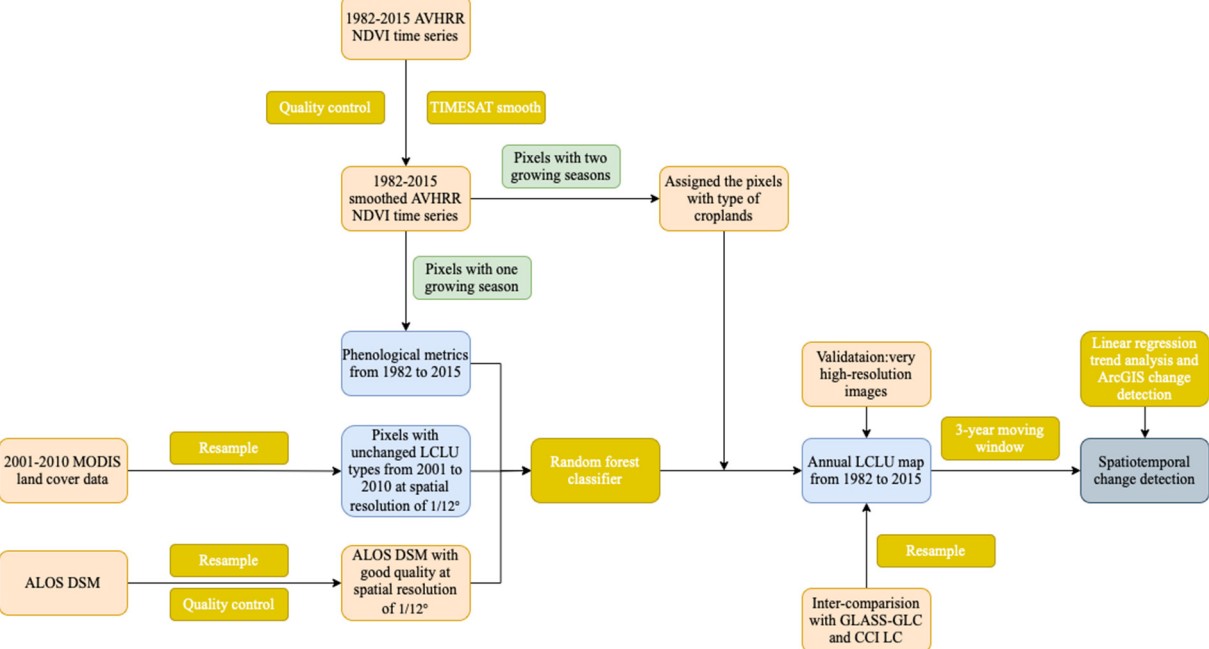

**Figure 2.** Flowchart of data pre-processing, random forest classification, post-processing, accuracy assessment, and change detection.

Owing to the possibility of unexpected transitions, such as croplands to urban areas and reversal to croplands within a short period, such transitions were excluded using a temporal filtering method ([29,51]). We adopted a 3-year moving window to test the consistency between the classes of year $n$ and $n + 2$, and if the class of year $n + 1$ was a disallowed transition as presented in Table 4, then this class was replaced using that of year $n$.

**Table 4.** Allowed and disallowed class transition for post-processing.

| | Class Number | Class | \multicolumn{13}{c}{Year $n + 1$} | | | | | | | | | | | | |
|---|---|---|---|---|---|---|---|---|---|---|---|---|---|---|---|
| | | | 1 | 2 | 3 | 4 | 5 | 8 | 9 | 10 | 12 | 13 | 14 | 16 | 17 |
| Year $n$ and $n + 2$ | 1 | Evergreen needleleaf forests | Yes | No | No | No | No | No | No | No | No | No | No | No | No |
| | 2 | Evergreen broadleaf forests | No | Yes | No | No | No | No | No | No | No | No | No | No | No |
| | 3 | Deciduous needleleaf forests | No | No | Yes | No | No | No | No | No | No | No | No | No | No |
| | 4 | Deciduous broadleaf forests | No | No | No | Yes | No | No | No | No | No | No | No | No | No |
| | 5 | Mixed forests | No | No | No | No | Yes | No | No | No | No | No | No | No | No |
| | 8 | Woody savannas | No | No | No | No | No | Yes | Yes | Yes | Yes | No | Yes | No | No |
| | 9 | Savannas | No | No | No | No | No | Yes | Yes | Yes | Yes | No | Yes | No | No |
| | 10 | Grasslands | No | No | No | No | No | No | No | Yes | Yes | No | Yes | Yes | No |
| | 12 | Croplands | No | No | No | No | No | No | No | Yes | Yes | No | Yes | Yes | No |
| | 13 | Urban and built-up lands | No | No | No | No | No | No | No | No | No | Yes | No | No | No |
| | 14 | Cropland/natural vegetation mosaic | No | No | No | No | No | No | No | Yes | Yes | No | Yes | No | No |
| | 16 | Barren | No | No | No | No | No | No | No | Yes | No | No | No | Yes | No |
| | 17 | Water bodies | No | No | No | No | No | No | No | No | No | No | No | No | Yes |

### 2.2.2. Accuracy Assessment

A traditional remote sensing accuracy assessment strategy was utilized to produce an error matrix based on 413 sample points generated using the very high-resolution Google Earth Pro images. We then converted the sample error matrix to a population error matrix, and calculated the user's, producer's, and overall accuracies according to a previous study [52]. The overall accuracy of the LCLU classification map for 2015 was then compared to those associated with MODIS and GLASS-GLC for the same year.

To inter-compare with the GLASS-GLC and CCI LC maps, a percentage of consistency was computed as follows:

$$\text{Consistency percentage} = \frac{N_{consistency}}{N} \times 100\% \tag{2}$$

where $N_{consistency}$ is the number of pixels with consistent LCLU types between the GLASS-GLC or CCI LC maps with those produced in the present study, while $N$ is the total number of pixels for the entire EASM region. The consistency percentage was calculated for each year.

### 2.2.3. Spatiotemporal Change Detection

Based on LCLU classification maps for the EASM region covering the period from 1982 to 2015, we performed spatiotemporal change detection to highlight changes in LCLU during the three decades. Areas representing forests, grasslands, croplands, urban and built-up, cropland/natural vegetation mosaic, barren, and water bodies were calculated. We then performed a linear regression trend analysis [44] with the area as dependent variable and time (i.e., 1982–2015) as independent variable for each LCLU type. We were interested in the slope of the linear regression to exhibit the temporal trend of LCLU areas. The significance of each trend was tested using the Student's *t*-test. In addition, we utilized the Change Detection tool in ArcGIS Pro 2.7 to explore LCLU transitions and their spatiotemporal changes.

A flowchart summarizing the data pre-processing, random forest classification, post-processing, accuracy assessment, and change detection is shown in Figure 2.

## 3. Results and Discussion

### 3.1. Performance of the Random Forest Classifier

The overall accuracy values for various ecological zones that were derived from 75% of the reference data are presented in Table 5. These values range from 70% to 98% for the nine zones, and these indicate an overall good performance of the random forest classifier

for the EASM region. The best performances of the classifier are associated with the temperate steppe (TS) and temperate desert (TD) zones as expected, because both zones are characterized by homogenous LCLU types (Figure 1b). The relatively poor performance of the random forests classifier for the subtropical humid forest (SH) zone is attributed to the mountainous landscape in southern China (Figure 1a). This observation is consistent with the findings of Zeng et al. (2019), who suggested that the classification of an area involving a complex topography or varying land cover types is characterized by low accuracy [53].

**Table 5.** Overall accuracy values of the random forest classifier for zones in the EASM region based on 75% of the reference data.

| | BC | BM | SH | SM | TS | TD | TC | TM | TP |
|---|---|---|---|---|---|---|---|---|---|
| Overall accuracy | 80% | 78% | 70% | 76% | 98% | 95% | 89% | 87% | 81% |

BC: Boreal coniferous forest; BM: Boreal mountain system; SH: Subtropical humid forest; SM: Subtropical mountain system; TS: Temperate steppe; TD: Temperate desert TC: Temperate continental forest TM: Temperate mountain system; TP: Tropical dry and moist forest, rainforest and mountain system.

### 3.2. Validation Using the Very High-Resolution Imagery

The overall accuracy (73%) of the 2015 classification map created in the present study is ~7% and 4% higher than those for the MODIS data and GLASS-GLC map, respectively, for the same year. These results further support the reliability of the map produced (Tables 6 and A1–A3). The user's and producer's accuracies vary from 55% to 97% and 1% to 95%, respectively. The higher accuracy values for the forests, grasslands, and barren land (>72%) are principally assigned to the associated relatively higher training data (Table 2), while the low producer's accuracy for water (5%) is mainly caused by the misclassification of water with grasslands and croplands (Table 6). The EASM region is heavily influenced by monsoon rainfall. The seasonal advancing and retreating of monsoon [54,55] may inundate some regions while drying up others, inducing the unexpected classification error of water bodies with other LCLU types. The poor producer's accuracy for the cropland/natural vegetation mosaic (1%) was primarily due to the fact that the majority of the training samples for this type were in the subtropical humid forest zone (Table 2), causing misclassification with forests and croplands. Similar phenological information of those classes, especially during the growing season, likely contributed to the misclassification [56]. It may also be due to the lower producer's accuracy of cropland/natural vegetation mosaic in MODIS land cover data that we used to produce reference data [40]. Regarding a coarse resolution, such as the 1/12° adopted in this study, urban areas are easily mixed with other LCLU types [57,58], such as small towns in the EASM region surrounded by croplands [59]. This explains the relatively poor producer's accuracy (15%) (Table 6).

**Table 6.** Validation for classified map in 2015 using very high-resolution imagery.

| | | Reference LCLU | | | | | | | | |
|---|---|---|---|---|---|---|---|---|---|---|
| | | Water Bodies | Forests | Grasslands | Croplands | Urban and Built-Up | Cropland/Natural Vegetation Mosaic | Barren | Total | User's Accuracy |
| Classified LCLU | Water bodies | 0.00046 | 0.00005 | 0.00005 | 0.00009 | 0.00005 | 0.00014 | 0.00000 | 0.00084 | 55% |
| | Forests | 0.00000 | 0.35816 | 0.02143 | 0.03061 | 0.00306 | 0.08571 | 0.00000 | 0.49897 | 72% |
| | Grasslands | 0.00278 | 0.00278 | 0.20279 | 0.02778 | 0.00556 | 0.00556 | 0.00833 | 0.25558 | 79% |
| | Croplands | 0.00480 | 0.01679 | 0.01679 | 0.09836 | 0.01679 | 0.01919 | 0.00000 | 0.17272 | 57% |
| | Urban and built-up | 0.00042 | 0.00000 | 0.00084 | 0.00084 | 0.00463 | 0.00084 | 0.00000 | 0.00757 | 61% |
| | Cropland/natural vegetation mosaic | 0.00000 | 0.00029 | 0.00000 | 0.00044 | 0.00015 | 0.00147 | 0.00000 | 0.00235 | 63% |
| | Barren | 0.00000 | 0.00000 | 0.00182 | 0.00000 | 0.00000 | 0.00000 | 0.06015 | 0.06197 | 97% |
| | Total | 0.00846 | 0.37807 | 0.24372 | 0.15812 | 0.03024 | 0.11291 | 0.06848 | 1 | |
| | Producer's accuracy | 5% | 95% | 83% | 62% | 15% | 1% | 88% | | 73% |

*3.3. Inter-Comparison of the LCLU Classification with the GLASS-GLC and CCI LC Products*

Classification maps produced in the present study exhibit percentages of consistency greater than 70% relative to those for the GLASS-GLC for all 34 years (Figure 3a). The minimum percentage of consistency (70.37%) between the LCLU classification and GLASS-GLC map emerged in 1983, while the maximum (77.62%) occurred in 2014. These discrepancies are mainly associated with the temperate mountain system, western boundary of the temperate steppe, northern border of the temperate continental forest, and subtropical humid forest zones. Conversely, consistency is common in the south of the temperate continental forest zone, including the North China Plain and the Korean Peninsula (Figures 1a and 4a). Spatial inconsistencies of LCLU types between the temperate steppe and subtropical humid forest zones were previously reported by [60]. Based on a comparison of five global land cover datasets involving China [60], it was demonstrated that these regions are vital for improvement of the overall LCLU data accuracy. The increasing correlations between the classified maps and the GLASS-GLC shown in Figure 3a are likely caused by the increasing consistency between the GIMMS NDVI data used in this study and the AVHRR NDVI data used in the GLASS-GLC product [61].

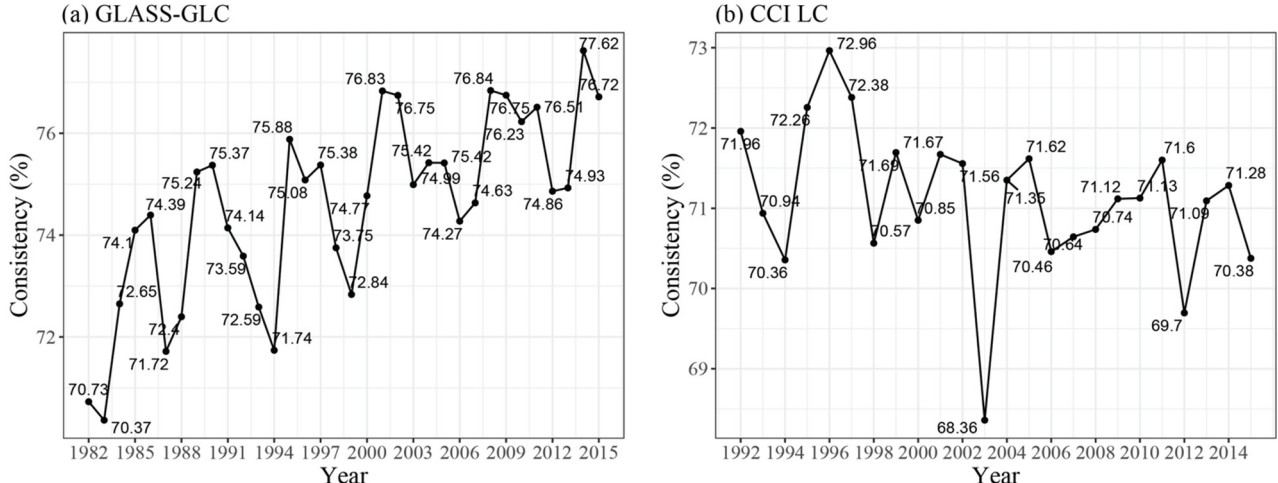

**Figure 3.** Percentage of consistency between the classification maps with (**a**) GLASS-GLC and (**b**) CCI LC.

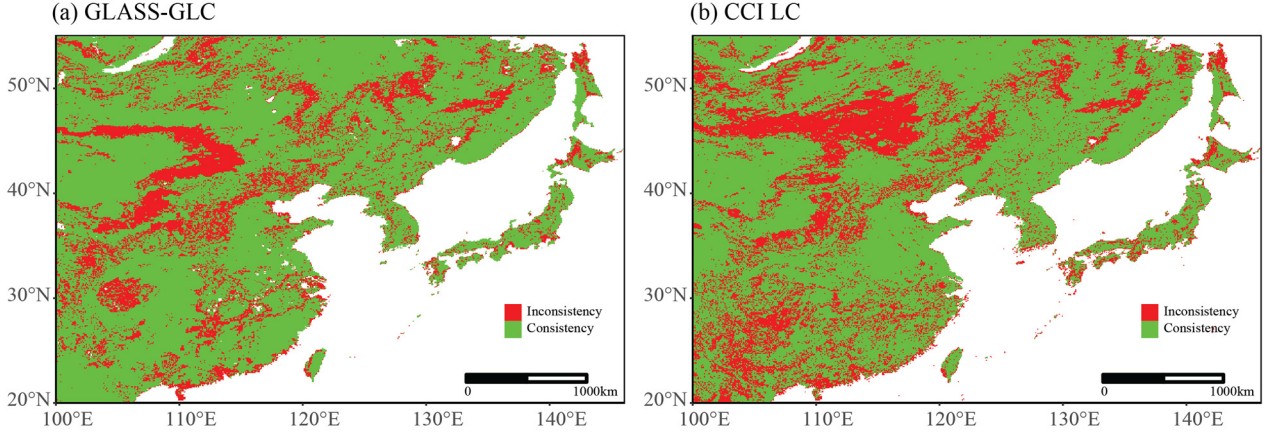

**Figure 4.** Spatial consistency between the classification map with (**a**) GLASS-GLC in 2014 and (**b**) CCI LC in 1996.

The spatial consistencies and inconsistencies between the classified maps and CCI LC maps are similar to those of GLASS-GLC maps, although the overall percentage of consis-

tency, which ranges from 68.36% to 72.96%, is lower for the later period (Figures 3b and 4b). The decreasing consistency values between the classification maps and the CCI LC product is attributed to the utilization of AVHRR NDVI data as input for CCI LC maps prior to the year 2000 [46]. The low consistency value of our classified map with CCI LC map in 2003 (Figure 3b) may be due to the relatively low consistency of GIMMS NDVI data used in this study and the SPOT-VGT NDVI used for generating CCI LC [62,63].

*3.4. Spatiotemporal LCLU Changes in the EASM Region*

Figure 5 shows the 1985, 1995, 2005, and 2015 LCLU maps for the EASM. Evidently, croplands are dominant in the North China Plain, Northeast China Plain, west of the Korean Peninsula, and south of Japan. These results are consistent with the distribution of paddy rice fields [64], which are generally known as breadbaskets in each country. Forests are common in the north of China, the east coast of Russia, the Korean Peninsula, and Japan. The main forest types in the EASM region are the deciduous broadleaf forests and mixed forests, and the minorly distributed evergreen broadleaf forests in southwestern China. Savannas are distributed in the north and south of the EASM, while barren lands surrounded by grasslands are frequent in Mongolia and Inner Mongolia. Urban and built-up lands are prevalent in cities including Shanghai, Guangzhou, and Beijing in China; Seoul in South Korea; and Nagoya, Osaka, and Tokyo in Japan. The spatial distributions of these LCLU types are consistent with those reported in previous studies [31,40,46,65].

According to the trend analysis, forests and croplands significantly increased in the EASM region from 1982 to 2015 at annual rates of $5260 \pm 2614$ km$^2$ and $3952 \pm 1334$ km$^2$, respectively. In contrast, grasslands significantly decreased by $7594 \pm 1732$ km$^2$/year (Figure 6). The urban and built-up, cropland/natural vegetation mosaic, and water bodies decreased at rates of $495 \pm 239$, $1536 \pm 437$, and $125 \pm 23$ km$^2$/year, respectively (Figure A3). Even though barren lands increased by approximately $394 \pm 634$ km$^2$/year, this trend was statistically insignificant (Figure A3). The increasing forests and croplands in the EASM region are consistent with findings based on country-level forest and CCI LC croplands data by [54]. Meanwhile, the LUH data revealed a decreasing trend for grasslands [66]. The decreasing trend in urban lands is mainly caused by classification errors linked to the coarse resolution, and this requires improvement in the future.

The analysis of spatiotemporal changes revealed that several LCLU transitions occurred in the EASM region during the period from 1985 to 2015 (Figure 7). In southern China, the transformation from croplands to forests represented the dominant transition for the period from 1985 to 1995. This transition was also prevalent from 1995 to 2005 and 2005 to 2015 (Figure 7b,c), and these trends are partially attributed to afforestation projects, such as the Grain for green in China [44]. Such projects probably also contributed to the cropland/natural vegetation mosaic to forests transition in southern China from 2005 to 2015 (Figure 7c). In contrast, croplands increased in north-eastern China mainly because of the grasslands to croplands (Figure 7a,c) and forests to croplands transitions (Figure 7b). These changes are mainly due to the increasing population and demand for food [60]. The demand for food, for instance, triggered an increase in the development of agriculture-related infrastructure, arable lands, and farms, and these enhanced the transition of grasslands to croplands in the Songnen Plain near Harbin in north-eastern China [67].

In the Amur River basin in Russia, which is on the north-eastern side of the Songnen Plain, croplands were predominantly converted to forests (savannas) between 1985 and 1995 (Figures 5b and 7a). In this region, the abandonment of arable lands during the 1990s was attributed to the collapse of the Soviet Union [68–71]. Conversely, the transition from forests (savannas) to croplands dominated from 1995 to 2005 and 2005 to 2015 in the region (Figures 5c,d and 7b,c). Recultivation was promoted by increasing the internal and external demand for crops, which were linked to expanding crop markets after China and Russia joined the World Trade Organization (WTO) in 2001 and 2012, respectively [71].

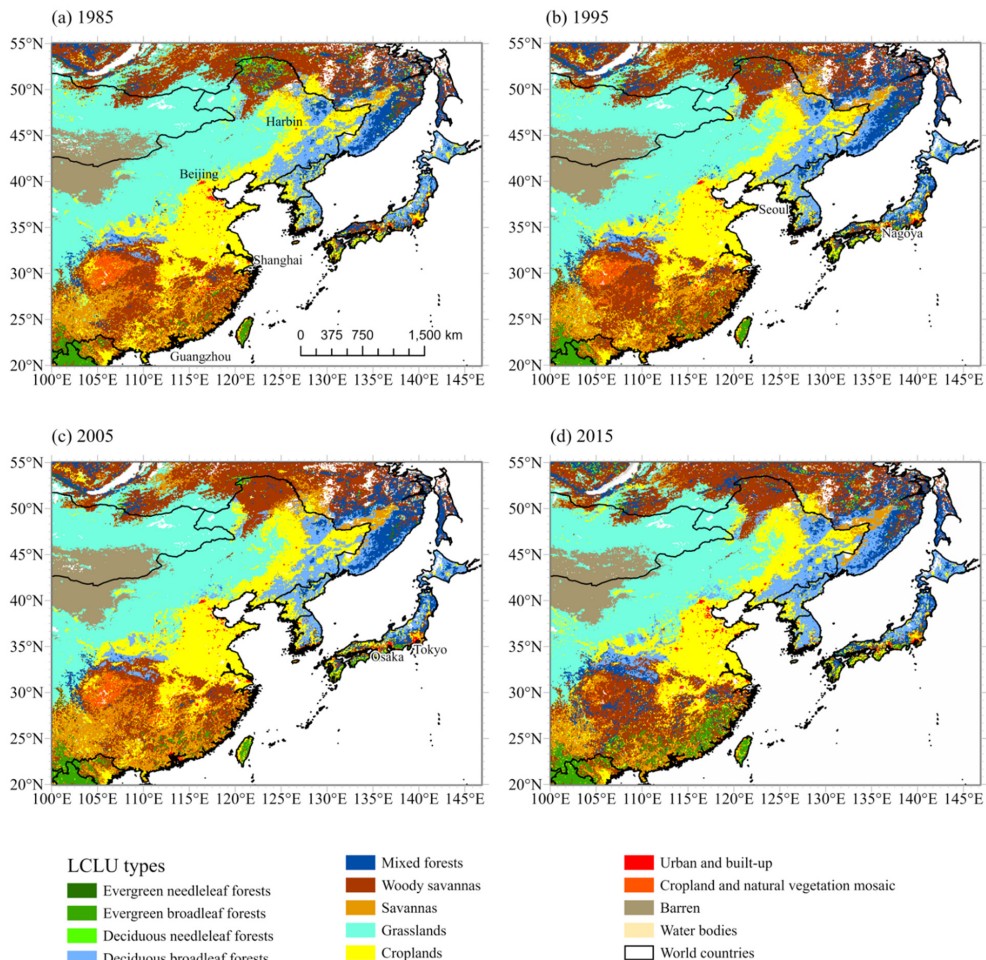

**Figure 5.** LCLU maps in the EASM region in (**a**) 1985; (**b**) 1995; (**c**) 2005; and (**d**) 2015.

In Mongolia and Inner Mongolia, the grasslands to barren land transition dominated from 1985 to 1995 and 1995 to 2005 (Figure 7a,b), whereas from 2005 to 2015, the main transition in these regions was the barren land to grasslands (Figure 7c). These LCLUC patterns are consistent with those reported in previous studies [72,73]. The grasslands to barren land transition from 1985 to 2005 was likely promoted by desertification associated with climate change and the increase in livestock [72]. The decreasing trend in precipitation in Mongolia during the 1990–2005 period was reversed after 2005 [73], and this contributed to the transition from barren land to grasslands observed from 2005 to 2015.

Regarding the Korean Peninsula, the croplands to forests transition dominated from 1985 to 1995 and 2005 to 2015 (Figure 7a,c), whereas from 1995 to 2005, the forests to croplands transition was prevalent (Figure 7b). The transition from forests to croplands was common in central North Korea from 1995 to 2005 (Figure 7b), and this was linked to deforestation due to the famine, economic, and energy crises in the late 1990s [74,75]. The transition from croplands to forests has dominated the Korean Peninsula (Figure 7c) since the year 2000 because of efforts to restore forests in North Korea. In contrast to North Korea, the reduction in croplands and forests due to urbanization reported in South Korea [75–77] was inconsistent with results in the present study, which revealed this was rather a minor transition (Figure 7). As in South Korea, croplands abandonment occurred in Japan due to changes in the structure of the population, such as aging and migration from rural areas [78–80]. This inconsistency may be linked to the inaccuracy in the classification of urban areas caused by the coarse resolution.

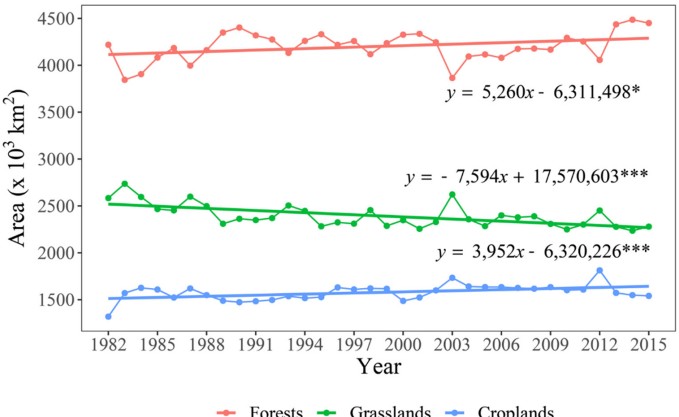

**Figure 6.** Temporal changes for the total areas of forests, croplands, and grasslands in the EASM region from 1982 to 2015 (The forests class includes the evergreen broadleaf, deciduous needleleaf, deciduous broadleaf, and mixed forests as well as woody savannas and savannas, as presented in Table 6. The unit of the change is km$^2$/year; while *, **, and *** represent the 10%, 5%, and 1% significance levels, respectively).

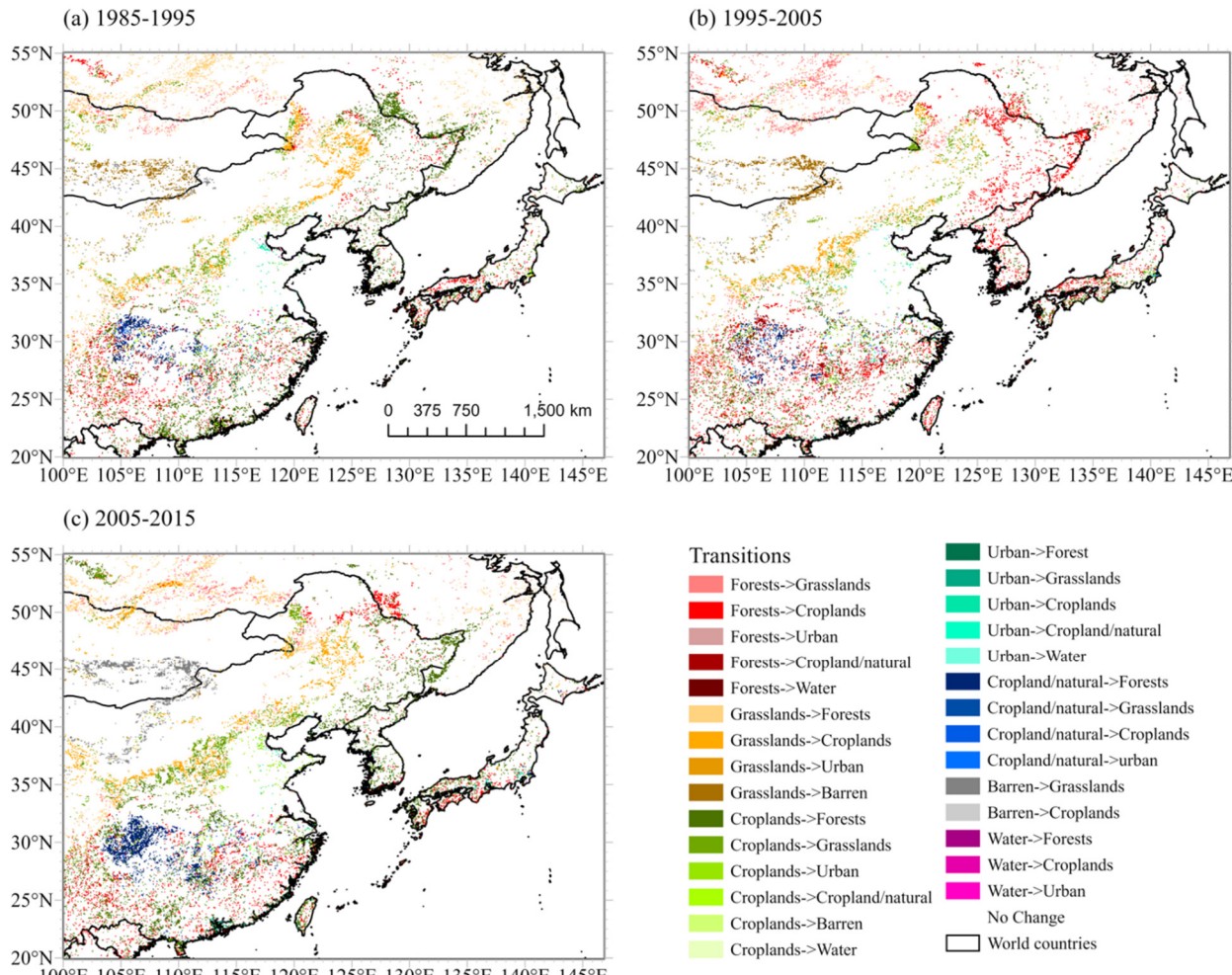

**Figure 7.** Transitions among the LCLU types in the EASM region for (**a**) 1985 to 1995, (**b**) 1995 to 2005, and (**c**) 2005 to 2015 (The forest class includes the evergreen broadleaf, deciduous needleleaf, deciduous broadleaf, and mixed forests as well as woody savannas and savannas, as presented in Table 6).

## 4. Conclusions

In the present study, a random forest classifier was utilized to produce annual scale, continuous, and 34-year LCLU maps for the EASM region from 1982 to 2015. These maps were created using phenological metrics derived from the GIMMS NDVI3g and elevation from the AW3D30 data. The 73% overall accuracy of the classification map for 2015 was 7% and 4% higher than those of MODIS and GLASS-GLC data for the same year. Moreover, the 34-year map displayed good spatial agreements with the GLASS-GLC and CCI LC products, captured increasing forest in southern China (Figure A2), and these confirmed the reliability of our classified maps.

LCLU detection analysis demonstrated that croplands and forests significantly increased in the EASM region during the period of 1982–2015. In contrast, the grasslands and cropland/natural vegetation mosaic decreased during the same period. The dominant LCLU transition in the EASM region over the last three decades was that of croplands to forests, and this was mainly linked to afforestation projects. Other transitions such as the grasslands to croplands also occurred. These explicit transitions provided insights into the land cover and land use management in the EASM region. In addition, the classification maps can be exploited as data sources for constraining the boundary conditions in Earth System Models (ESMs) and enhance understanding of variations in the EASM.

Nevertheless, the present study also involved limitations. First, the poor producer's accuracy for water bodies and the cropland/natural vegetation mosaic may propagate into the change detection process, introducing uncertainties to their spatiotemporal change patterns. To improve the classification accuracy for those two LCLU types, in our future study, other training data sources, such as that from field work, and additional ancillary data, such as precipitation data to represent monsoon advancing and retreating [81], and texture feature data to indicate different texture patterns of cropland/natural vegetation mosaic, forests, and croplands [82], will be included. Second, finer-resolution images (e.g., Landsat) may enhance the change detection for urban areas, which was also poorly quantified. Thirdly, this study did not explore the interactions of LCLUC and EASM, which needs to be addressed in our future study.

**Author Contributions:** Conceptualization, E.L.; methodology, J.O., Y.K. and Y.H.; software, J.O., Y.K. and Y.H.; validation, J.O. and Y.K.; formal analysis, J.O. and Y.K.; investigation, J.O., Y.K. and Y.H.; resources, E.L.; data curation, J.O. and Y.K.; writing—original draft preparation, Y.H.; writing—review and editing, J.O., Y.K., Y.H. and E.L.; visualization, J.O. and Y.H.; supervision, E.L. and Y.H.; project administration, E.L.; funding acquisition, E.L. All authors have read and agreed to the published version of the manuscript.

**Funding:** This study was supported by a grant from the National Research Foundation of Korea (NRF) funded by the Korean government (MSIT) (NRF-2020R1F1A1048886). Y.H. was partially supported by a University Research Council grant from the University of Central Arkansas.

**Institutional Review Board Statement:** Not applicable.

**Informed Consent Statement:** Not applicable.

**Data Availability Statement:** The GIMMS NDVI3g dataset is available through http://poles.tpdc.ac.cn/en/data/9775f2b4-7370-4e5e-a537-3482c9a83d88/ (accessed on 15 November 2020), the MODIS MCD12Q1 is downloaded from https://lpdaac.usgs.gov/products/mcd12q1v006/ (accessed on 26 April 2021), the shape file of ecological zones by FAO is from https://data.apps.fao.org/map/catalog/srv/eng/catalog.search;jsessionid=3699768EB0FA4370AE753A0A5B638D05?node=srv#/metadata/2fb209d0-fd34-4e5e-a3d8-a13c241eb61b (accessed on 22 July 2021), the AW3D30 is from https://portal.opentopography.org/raster?opentopoID=OTALOS.112016.4326.2 (accessed on 9 August 2021), GLASS-GLC is obtained from https://doi.pangaea.de/10.1594/PANGAEA.913496 (accessed on 8 January 2021), and ESA CCI LC is derived from https://www.esa-landcover-cci.org (accessed on 8 January 2021).

**Acknowledgments:** We thank the reviewers for their constructive comments to improve the manuscript.

**Conflicts of Interest:** The authors declare no conflict of interest.

## Nomenclature

| | |
|---|---|
| EASM | East Asian summer monsoon |
| LCLUC | Land cover and land use change |
| LCLU | Land cover and land use |
| MODIS | Moderate Resolution Imaging Spectroradiometer |
| GLASS | Global Land Surface Satellite |
| DSM | Digital Surface Model |
| ITCZ | Intertropical convergence zone |
| AMOC | Atlantic meridional overturning circulation |
| CMIP5 | Coupled Model Intercomparison Project Phase 5 |
| CMIP6 | Coupled Model Intercomparison Project Phase 6 |
| LUH | Land Use Harmonization |
| RegCM4 | 4th Regional Climate Model |
| WRF | Weather Research and Forecasting |
| AVHRR | Advanced Very High-Resolution Radiometer |
| GLASS-GLC | Global Land Surface Satellite-Global Land Cover |
| FROM-GLC_v2 | Finer resolution observation and monitoring of global land cover version 2 |
| GIMMS | Global Inventory Modeling and Mapping Studies |
| NDVI | Normalized Difference Vegetation Index |
| NDVI3g | Third generation GIMMS NDVI |
| ALOS | Advanced Land Observing Satellite |
| AW3D30 | ALOS World 3D—30m |
| CCI LC | Climate Change Initiative land cover |
| MCD12Q1 | MODIS land cover |
| ESA | European Space Agency |
| PRISM | Panchromatic remote-sensing instrument for stereo mapping |
| IGBP | International Geosphere-Biosphere Programme |
| UMD | University of Maryland |
| LAI | Leaf area index |
| CDRs | Climate data records |
| FAPAR | Fraction of absorbed photosynthetically active radiation |
| MERIS | Medium Resolution Imaging Spectrometer |
| SPOT-VGT | Système Probatoire d'Observation de la Terre Vegetation |
| FAO | Food and Agriculture Organization |
| BC | Boreal coniferous forest |
| BM | Boreal mountain system |
| SH | Subtropical humid forest |
| SM | Subtropical mountain system |
| TS | Temperate steppe |
| TD | Temperate desert |
| TC | Temperate continental forest |
| TM | Temperate mountain system |
| TP | Tropical dry and moist forest, rainforest and mountain system |
| WTO | World Trade Organization |

## Appendix A

**Table A1.** Validation for MODIS LCLU in 2015 using very high-resolution imagery.

| | | Reference LCLU | | | | | | | | |
|---|---|---|---|---|---|---|---|---|---|---|
| | | Water Bodies | Forests | Grasslands | Croplands | Urban and Built-Up | Cropland/Natural Vegetation Mosaic | Barren | Total | User's Accuracy |
| **MODIS LCLU** | **Water bodies** | 0.03193 | 0.00456 | 0.00456 | 0.00456 | 0.00000 | 0.01369 | 0.00000 | 0.05930 | 54% |
| | **Forests** | 0.00872 | 0.33432 | 0.02907 | 0.03489 | 0.00291 | 0.09012 | 0.00000 | 0.50002 | 67% |
| | **Grasslands** | 0.00295 | 0.01182 | 0.18609 | 0.03249 | 0.00295 | 0.00591 | 0.01182 | 0.25402 | 73% |
| | **Croplands** | 0.00667 | 0.00667 | 0.02447 | 0.07563 | 0.01557 | 0.01335 | 0.00000 | 0.14237 | 53% |
| | **Urban and built-up** | 0.00000 | 0.00082 | 0.00246 | 0.00410 | 0.01065 | 0.00246 | 0.00000 | 0.02049 | 52% |
| | **Cropland/natural vegetation mosaic** | 0.00000 | 0.00143 | 0.00000 | 0.00286 | 0.00000 | 0.00786 | 0.00000 | 0.01215 | 65% |
| | **Barren** | 0.00000 | 0.00032 | 0.00097 | 0.00000 | 0.00000 | 0.00000 | 0.01035 | 0.01165 | 89% |
| | **Total** | 0.05028 | 0.35994 | 0.24762 | 0.15453 | 0.03209 | 0.13338 | 0.02217 | 1.00000 | |
| | **Producer's accuracy** | 64% | 93% | 75% | 49% | 33% | 6% | 47% | | 66% |

**Table A2.** Validation for classified map in 2015 using very high-resolution imagery based on four classes.

| | | Reference LCLU | | | | | |
|---|---|---|---|---|---|---|---|
| | | **Forests** | **Grasslands** | **Croplands** | **Barren** | **Total** | **User's Accuracy** |
| **Classified LCLU** | **Forests** | 0.36256 | 0.02188 | 0.11877 | 0.00000 | 0.50321 | 72% |
| | **Grasslands** | 0.00290 | 0.21140 | 0.03475 | 0.00869 | 0.25774 | 82% |
| | **Croplands** | 0.01859 | 0.01162 | 0.14636 | 0.00000 | 0.17656 | 83% |
| | **Barren** | 0.00000 | 0.00184 | 0.00000 | 0.06065 | 0.06249 | 97% |
| | **Total** | 0.38404 | 0.24673 | 0.29988 | 0.06934 | 1.00000 | |
| | **Producer's accuracy** | 94% | 86% | 49% | 87% | | 78% |

Note the four classes are determined by the common classes between our classified map and GLASS-GLC, as indicated in Table 3.

**Table A3.** Validation for GLASS-GLC LCLU in 2015 using very high-resolution imagery based on four classes.

| | | Reference LCLU | | | | | |
|---|---|---|---|---|---|---|---|
| | | **Forests** | **Grasslands** | **Croplands** | **Barren** | **Total** | **User's Accuracy** |
| **GLASS-GLC LCLU** | **Forests** | 0.34935 | 0.03949 | 0.10329 | 0.00000 | 0.49213 | 71% |
| | **Grasslands** | 0.00687 | 0.16140 | 0.04121 | 0.00000 | 0.20947 | 77% |
| | **Croplands** | 0.01713 | 0.02783 | 0.14346 | 0.00000 | 0.18842 | 76% |
| | **Barren** | 0.00000 | 0.02918 | 0.00000 | 0.08080 | 0.10998 | 73% |
| | **Total** | 0.37335 | 0.25790 | 0.28795 | 0.08080 | 1.00000 | |
| | **Producer's accuracy** | 94% | 63% | 50% | 100.00% | | 74% |

Note the four classes are determined by the common classes between our classified map and GLASS-GLC, as indicated in Table 3.

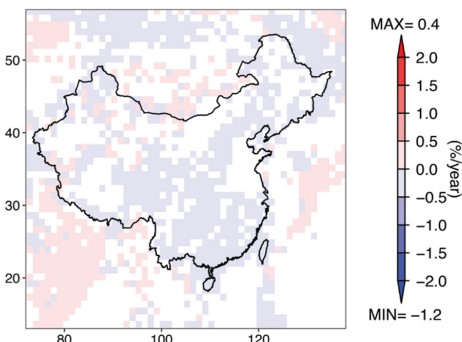

**Figure A1.** Temporal trend of forest fraction from 1982 to 2013 derived from LUH dataset (color bar is the slope in percentage per year).

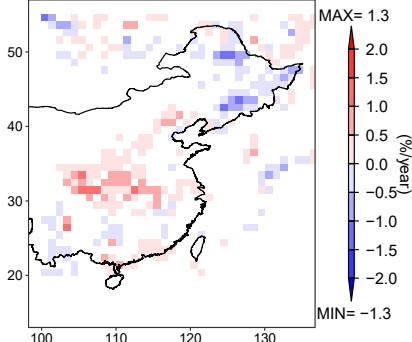

**Figure A2.** Temporal trend of forest fraction from 1982 to 2013 derived from our classified maps (color bar is the slope in percentage per year).

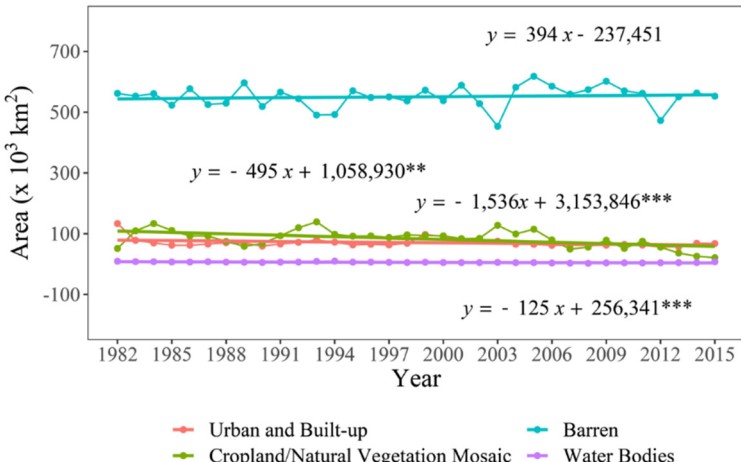

**Figure A3.** Temporal changes for the total areas of urban and built-up, cropland/natural vegetation mosaic, barren, and water bodies in the EASM region from 1982 to 2015 (The forest class includes evergreen broadleaf, deciduous needleleaf, deciduous broadleaf, and mixed forests as well as woody savanna and savanna, as presented in Table 6; The unit of the change is $km^2$/year; *, **, and *** represent 10%, 5%, and 1% significant level, respectively).

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
