# Peer review of "Land Cover and Land Use Mapping of the East Asian Summer Monsoon Region from 1982 to 2015"

_land, doi:10.3390/land11030391_

Round 1
Reviewer 1 Report
The manuscript uses several analytical methods to analyse changes in land use and land cover (LULC) in East Asia Summer Monsoon region (EASM). Changes in land use and land cover is well studied at global and various regional scale1, 2,3. A fact that the authors have acknowledged. Despite the attempt to justify the need for LULC impact on summer monsoon in East Asia. There are several studies that have assessed the feedback of land use change on EASM4,5. Authors identify the need to “…enhanced the quantification of LULC effects on EASM” as a justification for this study; however, there is no evidence of this enhanced quantification in the results and discussions. Consequently, there is no new insight between LULC and its impact on summer Monsoon. Authors used several methods (e.g., TIMESAT and Random Forest) without clear connection between the methods and associated outputs.
1Potapov, P., Hansen, M.C., Kommareddy, I., Kommareddy, A., Turubanova, S., Pickens, A., Adusei, B., Tyukavina, A. and Ying, Q., 2020. Landsat analysis ready data for global land cover and land cover change mapping. Remote Sensing, 12(3), p.426.
2Song, X.P., Hansen, M.C., Stehman, S.V., Potapov, P.V., Tyukavina, A., Vermote, E.F. and Townshend, J.R., 2018. Global land change from 1982 to 2016. Nature, 560(7720), pp.639-643.
3Hansen, M.C., Wang, L., Song, X.P., Tyukavina, A., Turubanova, S., Potapov, P.V. and Stehman, S.V., 2020. The fate of tropical forest fragments. Science Advances, 6(11), p.eaax8574.
4Cao, F., Dan, L., Ma, Z. and Gao, T., 2020. Assessing the regional climate impact on terrestrial ecosystem over East Asia using coupled models with land use and land cover forcing during 1980–2010. Scientific Reports, 10(1), pp.1-15.
5Niu, X., Tang, J., Wang, S. and Fu, C., 2019. Impact of future land use and land cover change on temperature projections over East Asia. Climate dynamics, 52(11), pp.6475-6490.
L14. Why controversial?
L24 – 26 In what ways can the understanding of land-atmosphere interactions be enhanced?
L37 – 38 This sentence is very confusing sentence!
L52 – 57 The preceding sentence/statements does not support the assertion of the authors that “The inconclusive finding of LCLUC-EASM interaction is possibly because of the scarcity of temporally continuous LCLU data for this region.”
L80 – 81 Was the validation dataset based on ground-truth data?
L109 – 128 It is unclear how the seasonal parameters translate to land surface classification (see, line 128). TIMESAT is designed to model structural and functional changes in different ecosystems/land cover types. This is done by removing any bias towards high NDVI values often associated with large-sized land cover types (e.g., forest) and modelling seasonal peaks in the landscape. However, the authors have mentioned that the objective is to produce annual LCLU. Considering the modelling bias, how does this method fit with the objective of this study? What will the land surface be further classified into?
L131 – 195 Not sure what the purpose is here. Why is it necessary to implement a secondary classification using DSM? Why is MODIS land cover dataset useful? Authors already plan to classify the land surface using NDVI dataset (in TIMESAT). Why was the ICGP classification scheme selected? Authors mentioned that it has 17 classes. Why is the number of classes relevant?
L200 Why use one growing season?
Table 6 Number of decimal places suggest very high precision. Is the validation exercise that precise? In addition, the overall classification accuracy of 73% is not very high.
Figure 3 What is the consistency based on? The accuracy assessment? Similarity or dissimilarity in classification?
Author Response
A point-by-point response to all the reviewer’s comments is attached here.
The manuscript with tracked changes is uploaded along with a revised manuscript.

Reviewer 2 Report
The only remark deals with the oversaturation of the text of the manuscript with the abbreviations. Reviewer understands the authors used to this abbreviations in their everyday practice, but the authors also must understand the readers, who are not the professionals in the authors field of study. The results of study must be clear for readers who are the specialists in other fields of science. From the reviewers point of view the abbreviations can not be used in the abstract and in the keywords. It will be better, if there will be special list of abbreviations between, e.g. Abstract and Introduction. The manuscript will be better from it. The paper is an good example of rather successfull study of some ecological shifts in the studied area.
Author Response

(The authors gave the same response as above.)

Reviewer 3 Report
This research is using AVHRR GIMS data to classify land use/land cover in East Asian summer monsoon region and to compare its accuracy to MODIS and GLASS land products.
The research is generally well structured and written, but I have several concerns that should be addressed:
Abstract Line 17 "This was achieved..." - I think that you should add information that map was created using AVHRR GIMS because it is little unclear.
Chapter 2.2.3 /Figure 2. - There is no explanation on linear regression and parameters used. Please add and explain all steps of processing, pre- and post-processing.
Line 284-286 "The poor producer's..." - This is very bad result. With this result you should either choose new samples, or combine this class with other classes.
Why did you choose cropland and cropland/natural vegetation because you should expect this to happen?
This should be corrected or the methodology should be corrected.
Table 6. - How do you explain low accuracy for water bodies and "urban and built-up" classes? Those classes usually have high accuracy.
Figure 3 - How do you explain decrease in consistency in 2003 for CCI LC?
Figure 6 - Figure is not referenced/mentioned in text.
Author Response

(The authors gave the same response as above.)

Round 2
Reviewer 3 Report
It is much better and ready for publication.